# A Novel Quality Control Method for the Determination of the Refractive Index of Oil-in-Water Creams and Its Correlation with Skin Hydration

Deborah Adefunke Adejokun and Kalliopi Dodou *

School of Pharmacy and Pharmaceutical Sciences, Faculty of Health Sciences and Wellbeing, University of Sunderland, Sunderland SR1 3SD, UK; bg69bo@research.sunderland.ac.uk
* Correspondence: kalliopi.dodou@sunderland.ac.uk; Tel.: +44-(0)191-515-2503

**Abstract:** The sensory properties of cosmetic products can influence consumers' choice. The accurate correlation of sensory properties, such as skin hydration, with the material properties of the formulation could be desirable. In this study, we aimed to demonstrate a new method for the in vitro measurement of the refractive indices (RIs) of turbid creams. The critical wavelength of each cream was obtained through direct measurement using a sun protection factor (SPF) meter; the wavelength value was then applied in the Sellmeier equation to determine the RI. The results obtained from the in vitro skin hydration measurement for each cream correlated with their RI values. This suggests that RI measurements could be a useful predictive tool for the ranking of creams in terms of their skin hydration effects.

**Keywords:** sensory testing; refractive index; critical wavelength; turbidity; skin hydration; creams





## 1. Introduction

Sensory analysis plays a crucial role in the field of cosmetic science. It is used for claim substantiation via subjective users' perception also providing an understanding on how sensory attributes influence consumer's choice and, in turn, the market success of the product [1]. The Organization of Standardization (ISO) allows the properties of a cosmetic product to be described using both qualitative and quantitative methods [2]. This is performed by the selection of a plain descriptive lexicon and a team of well-trained judges to qualify and quantify the test products on the basis of their individual sensory perception via scoring each attribute on a given scale. The statistical evaluation of the collected data can then correlate the scores, assess the overall performance of the product and derive valid claims [3,4]. However, user trials involving scoring scales can be biased due to the subjective nature of the collected data [5,6]. Therefore, there is a need for the development of instrumental techniques that can reliably correlate to and predict sensory properties [7].

In a previous study, we demonstrated how sensory attributes of semisolids, such as pourability, firmness, elasticity, spreadability and stickiness, can be accurately correlated to the rheological measurements of the formulation [7]. In this study, we aimed to develop further such correlations.

The refractive index (RI) of a material is a measure of the velocity of light in vacuum divided by the velocity of light crossing the material. The RI has a wide range of applications, from the estimation of drug concentration present in a sample to the opaqueness or turbidity of the sample [8]. Changes in the RI of the skin have been shown to correlate to skin hydration after the application of a moisturizer; this is because hydration renders the skin less opaque, i.e., more translucent, resulting in a decrease in the RI [9–11]. The measurements of the RIs of cosmetic creams could potentially be correlated to their short-term skin hydration effects, assuming that less turbid creams with a low RI will

have a high water content or/and dissolved actives with a hydrating effect. Because of the turbid nature of creams, a traditional UV spectrophotometer cannot be used to measure their RIs [12,13], and the optical coherence tomography technique cannot elaborate their RIs on a routine basis [9]; therefore, there is a need for developing a simple method to accurately and routinely measure the RIs of creams and turbid samples.

The objectives of the present study were as following: (i) to develop a method for the direct and accurate measurement of the RIs of turbid formulations; and (ii) to investigate if there is a correlation between the RIs of creams and their skin hydration effects.

## 2. Materials and Methods

### 2.1. Materials

The active ingredient (X), cholesterol, span65 and solutol HS-15 were obtained from Sigma-Aldrich, Inc. (Gillingham, UK). Baobab oil was purchased from Aromatic Natural Skin Care (Forres, UK), and jojoba and coconut oil were bought from SouthernCross Botanicals (Knockrow, Australia). The Emulsifying Wax was obtained from CRODA International Plc (Goole, East Yorkshire, UK). Other excipients of the cream and Tris buffer solutions were of analytical grade.

### 2.2. Methods

#### 2.2.1. Preparation of the Oil-in-Water Creams

Oil-in-water creams IA–IVA and IB–IVB used in this study were formulated as explained in our previous paper [7].

Four active-containing oil-in-water creams (labelled A) and their controls (labelled B; without active ingredients) were prepared. Each cream contained the following oil combinations: I (volume ratio of 8% jojoba and baobab oils, 1:1)—water phase (85%), oil phase (10%) and emulsifier (5%); II (volume ratio of 10% jojoba and baobab oil, 1:1), III (volume ratio of 10% jojoba oil and coconut oil, 1:1) and IV (volume ratio of 10% baobab and coconut oil, 1:1)—water phase (83%), oil phase (12%) and emulsifier (5%). The composition of each cream is shown in Table 1.

**Table 1.** Ingredients and % (*w/w*) composition of each cream formulation.

| Cream Phases | Ingredients | Composition of Each Cream Formulation (% *w/w*) | | | | | | | |
|---|---|---|---|---|---|---|---|---|---|
| | | IA | IB | IIA | IIB | IIIA | IIIB | IVA | IVB |
| Oil phase | Stearyl alcohol | | | | | 1 | 1 | 1 | 1 |
| | Jojoba oil | 4 | 4 | 5 | 5 | 5 | 5 | | |
| | Baobab oil | 4 | 4 | 5 | 5 | | | 5 | 5 |
| | Coconut oil | | | | | 5 | 5 | 5 | 5 |
| Water phase | Glycerine | 5 | 5 | | | 5 | 5 | 5 | 5 |
| | Propylene glycol | | | 5 | 5 | | | | |
| | Water | 73.7 | 78.7 | 71.7 | 76.7 | 71.7 | 76.7 | 71.7 | 76.7 |
| Active | Entrapped active ingredient | 5 | | 5 | | 5 | | 5 | |

#### 2.2.2. Measurement of Refractive Index Using an SPF Analyser

A sun protection factor (SPF) analyser, SPF-290AS (SolarLight®, Glenside, PA, USA), was used to obtain the individual transmittance wavelength of each cream sample. The RI of each cream was then calculated using the Sellmeier equation (Equation (1)):

$$n^2(\lambda) = 1 + (B_1\lambda^2/\lambda^2 - C_1) + (B_2\lambda^2/\lambda^2 - C_2) + (B_3\lambda^2/\lambda^2 - C_3), \quad (1)$$

where

n is the refractive index (RI),

λ is the wavelength of the test sample Determined by a SPF meter.

The coefficients of the Sellmeier equation for a fused silica/silicon substrate can be shown as following [14]:

$B_1$ = 0.696166300; $C_1$ = 4.67914826 × $10^{-3}$ $\mu m^2$;

$B_2$ = 0.407942600; $C_2$ = 1.35120631 × $10^{-2}$ $\mu m^2$;

$B_3$ = 0.897479400; $C_3$ = 97.9340025 $\mu m^2$.

A wavelength measurement on the SPF working at the wavelength of 290 m was carried out by initially placing an empty silicon substrate or a transpore tape (to imitate the skin surface) in the optical path to acquire a reference scan. The substrate was then loaded with the test sample at 2.0 $\mu L/cm^2$, spread out in a unidirectional motion, allowed to dry for 15 min and returned to the optical path. Six different scans were taken by a monochromator over the wavelength region of 380–500 nm, and an average scan was produced. The SPF software factors out the reference scan data, resulting in the transmittance of only the measured sample. This experiment was conducted in triplicate for each cream at room temperature.

### 2.2.3. Skin Hydration

A preliminary self-evaluation of the hydrating effects of the creams was carried out by the researcher using a skin hydration meter (Moisture meter SC Compact, Delfin, UK). Hands were washed with a soap and dried for 5 min. The moisture meter was placed on the back of the hand and held in a steady position, until a measurement was taken at T0 (t = 0) and the value was recorded. A small amount of cream was then applied in a circular motion to the same location of the original reading, and after 5 min, a second measurement was taken at T5 (t = 5). This process was repeated in triplicate for each cream.

### 2.2.4. Statistical Analysis

A statistical evaluation of results was carried out using the IBM SPSS software. To indicate whether any significant difference ($p < 0.05$) existed in the skin hydration after the cream application, a paired sample *t*-test was used to compare the results before and after measurements.

### 3. Results and Discussion

*Correlation between RI Measurements and Skin Hydration Results*

Table 2 shows the calculated RI values for all cream models. All creams were opaque or turbid with RI values greater than the RI of full fat milk cream, which is 1.38810 [12]. The least turbid creams were models IA and IB, having the lowest RI values (Table 2).

**Table 2.** The critical wavelengths of all models taken in triplicate and their refractive index (RI) values.

| Model | Wavelength 1 | Wavelength 2 | Wavelength 3 | Mean Wavelength (nm) | ±SD | Wavelength (μm) | RI Value |
|---|---|---|---|---|---|---|---|
| IA | 387.7 | 387.8 | 387.8 | 387.8 | 0.06 | 0.3878 | 2.12377 |
| IB | 387.7 | 387.7 | 387.8 | 387.7 | 0.06 | 0.3877 | 2.12378 |
| IIA | 385.0 | 384.9 | 385.0 | 385.0 | 0.06 | 0.3850 | 2.12397 |
| IIB | 385.1 | 385.0 | 385.0 | 385.0 | 0.06 | 0.3850 | 2.12397 |
| IIIA | 385.4 | 385.4 | 385.4 | 385.4 | 0 | 0.3854 | 2.12393 |
| IIIB | 385.5 | 385.6 | 385.5 | 385.5 | 0.06 | 0.3855 | 2.12393 |
| IVA | 385.9 | 385.5 | 385.9 | 385.8 | 0.23 | 0.3858 | 2.12391 |
| IVB | 385.9 | 385.9 | 385.9 | 385.9 | 0 | 0.3859 | 2.12392 |

Table 3 shows the % increase in skin hydration after the application of each cream. The paired sample *t*-test revealed there was a statistically significant difference in skin hydration (*p*-value < 0.05) before and after the measurements of skin hydration for most cream models.

**Table 3.** Measurement values of skin hydration in percentage (%) before and after the measurements of skin hydration ($T_0$ vs. $T_5$).

| Cream Model/ Number | 1 | | 2 | | 3 | | Mean | | | % Increase in Hydration |
|---|---|---|---|---|---|---|---|---|---|---|
| | Before ($T_0$) | After ($T_5$) | Before ($T_0$) | After ($T_5$) | Before ($T_0$) | After ($T_5$) | Before ($T_0$/±SD) | After ($T_5$/±SD) | *p*-Value | |
| IA/1 | 51.2 | 59.4 | 29.6 | 55.9 | 55.5 | 59.8 | 45.4/±13.9 | 58.4/±2.1 | 0.09 | 28.63 |
| IB/2 | 36.4 | 52.3 | 59.8 | 67.2 | 51.7 | 65.3 | 49.3/±11.9 | 61.6/±8.1 | 0.02 | 24.94 |
| IIA/3 | 58.7 | 62.4 | 58.0 | 67.4 | 51.5 | 55.6 | 56.1/±4.0 | 61.8/±5.9 | 0.04 | 10.16 |
| IIB/4 | 44.5 | 57.1 | 66.8 | 67.2 | 48.0 | 58.7 | 53.1/±11.9 | 61.0/±5.4 | 0.08 | 14.87 |
| IIIA/5 | 67.4 | 72.0 | 60.3 | 68.6 | 54.9 | 63.8 | 60.9/±6.3 | 68.1/±4.1 | 0.01 | 11.82 |
| IIIB/6 | 65.2 | 65.7 | 70.5 | 77.1 | 62.1 | 73.0 | 65.9/±4.2 | 71.9/±5.7 | 0.09 | 9.10 |
| IVA/7 | 58.2 | 64.7 | 55.6 | 60.8 | 65.9 | 70.5 | 59.9/±5.4 | 65.3/±4.9 | 0.005 | 9.01 |
| IVB/8 | 61.0 | 74.1 | 66.4 | 73.9 | 59.4 | 66.5 | 62.3/±3.7 | 71.5/±4.3 | 0.02 | 14.76 |

According to these preliminary results, creams IA and IB showed the highest increase in hydration alongside the lowest RI values, therefore confirming the hypothesis that creams with a lower RI have a higher hydrating effect. This correlation is depicted in Figure 1.

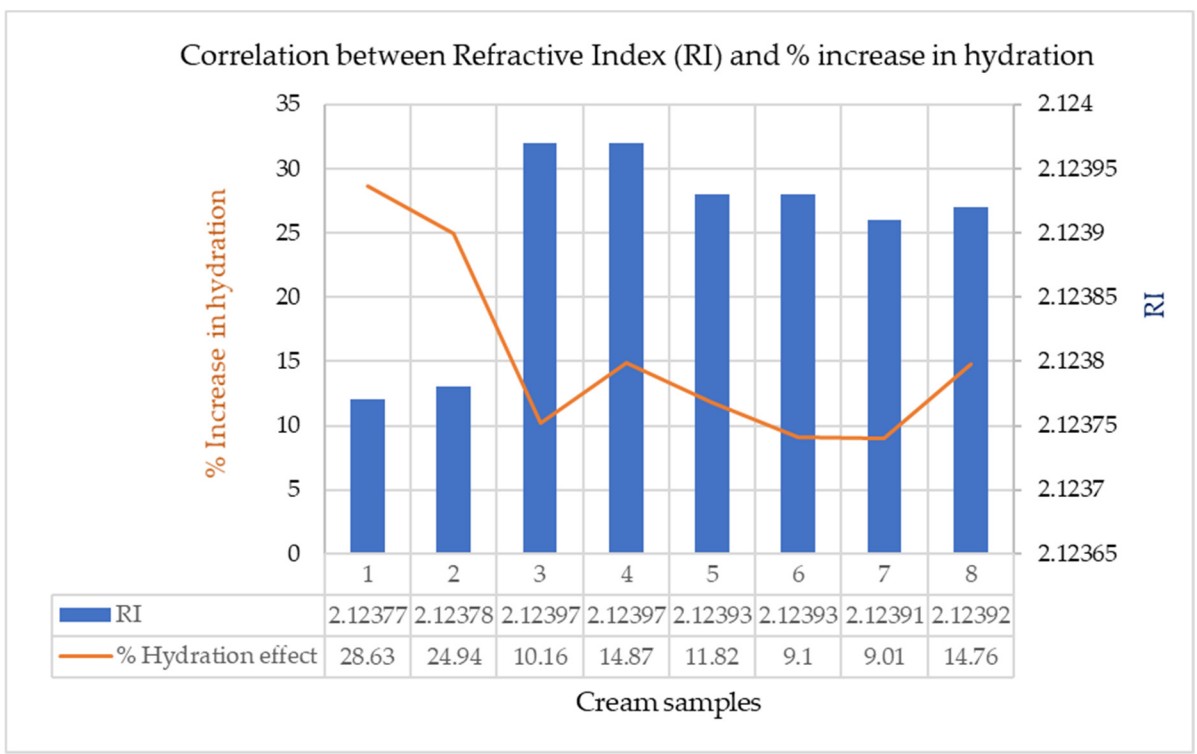

**Figure 1.** Relationships between RI and % increase in hydration. Cream samples (IA to IVB) are shown in numerical order of 1–8.

The high skin hydration effect seen in models IA and IB could be attributed to the higher water contents in their compositions compared to the rest of the creams (Table 1). This effect can also be attributed to the humectant used and its compatibility with jojoba

and baobab oil [15], considering that cream models IIA and IIB contained the same type of oils as IA and IB but with a different humectant. In fact, the turbid nature and low hydration effect of models IIA and IIB could be the effect of propylene glycol (Table 1) in comparison to the glycerin present in IA and IB. Based on these observations, the water content and the type of humectant seemed to be the determining factors in the hydration effect. Changes in these parameters can be detected by RI measurements, via changes in ingredients' solubility/compatibility, therefore explaining the observed correlation between RI/turbidity and hydration effects. Similarly, the high RI and the low hydration of models III and IV can be attributed to the presence of stearyl alcohol in models III and IV (Tables 1–3) and its incompatibility with glycerine and/or the oils in these formulas.

## 4. Conclusions

In this study, we reported our preliminary findings on a new method for the prediction of the skin hydration effects of creams by measuring their RIs, using the Sun Protection Factor-290 Automated System (SPF-290AS). This newly developed method using the SPF equipment and the Sellmeier equation is highly sensitive and simple and allows the determination of the RIs of turbid samples without dilution. The correlation of the RIs of the creams with their skin hydration effects could be a predictive tool for the cosmetics industry to provide useful information on the hydration effect of creams before embarking on user trials and/or confirming the results from user trials. Further studies should include a large-scale user trial to validate this predictive tool, also including different types of semisolid formulations (water-in-oil and gels) to explore the extent of its applicability.

**Author Contributions:** Conceptualization, D.A.A. and K.D.; methodology, D.A.A.; validation, D.A.A. and K.D.; formal analysis, D.A.A. and K.D.; investigation, D.A.A.; resources, K.D.; data curation, D.A.A.; writing—original draft preparation, D.A.A.; writing—review and editing, K.D.; supervision, K.D.; project administration, K.D. All authors have read and agreed to the published version of the manuscript.

**Funding:** This research received no external funding.

**Institutional Review Board Statement:** The study was conducted according to the guidelines of the Declaration of Helsinki, and approved by the Institutional Review Board (or Ethics Committee) of The University of Sunderland (protocol code 006180 and date of approval: 20 February 2020).

**Informed Consent Statement:** Informed consent was obtained from all subjects involved in the study.

**Data Availability Statement:** Data is contained within the article.

**Conflicts of Interest:** The authors declare no conflict of interest.

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
