# Peer review of "A Novel Quality Control Method for the Determination of the Refractive Index of Oil-in-Water Creams and Its Correlation with Skin Hydration"

_cosmetics, doi:10.3390/cosmetics8030074_

Round 1

Reviewer 1 Report

Cosmetics-1297907-peer-review

Comments:

The present study shows a promising research about a developped method for the accurate measurement of the RI of turbid matrices and to verify the hypothesis that there is a correlation of the Ri of the creams with their skin hydration effect.

In general, the manuscript is more like a short communication; methods are correct, but it is necessary to detail the samples they worked with; results and discussion is too concise, and leaves some uncertainty in the reader.

  Some suggestions to improve the quality of the work are the following:

  • Line 13: –SPF- The abbreviation should be avoided in the abstract. If you do use it, its meaning should also be written there (similar to RI).
  • Line 24-25: The reference [2] the authors used is not the best choice. Why don’t use the ISO standard number for that? : ISO 13299:2016
  • Line 56-57: An accurate and detailed description of the samples used is essential to facilitate understanding of the results (to give only a reference to these is not sufficient)
  • Line 62 and 76: I miss/suggest to add the source designation for the Sellmeier equation and coefficients the authors used
  • Line 81-82: It is questionable to the interpretation of the results that only one person’s skin has been studied for the hydration effect. Because of it, I suggest a little modification in the Results and Discussion section (e.g.: preliminary results …)
  • The differences of the RI values of the samples are small (Table 1). If the authors use different ingredients in the samples, it will also involve a change in RI values. But, if you change more than one parameter in a sample whether we can be sure that there is still a correlation between RI and the hydration effect.
  • Line 118: "determination of the IR of..." to be corrected: RI
  • The relationship between the two parameters (RI and hydration effect %) could be illustrated in a figure which would improve the illustration of the results.

Author Response

We would like to thank the Reviewer for their constructive feedback.

Comments:

The present study shows a promising research about a developed method for the accurate measurement of the RI of turbid matrices and to verify the hypothesis that there is a correlation of the Ri of the creams with their skin hydration effect.

In general, the manuscript is more like a short communication; methods are correct, but it is necessary to detail the samples they worked with; results and discussion is too concise, and leaves some uncertainty in the reader.

  Some suggestions to improve the quality of the work are the following:

  • Line 13: –SPF- The abbreviation should be avoided in the abstract. If you do use it, its meaning should also be written there (similar to RI).

Done

  • Line 24-25: The reference [2] the authors used is not the best choice. Why don’t use the ISO standard number for that? : ISO 13299:2016

Thank you, very good point. Reference [2] is now the ISO standard. The numbering of the subsequent references has been revised.

  • Line 56-57: An accurate and detailed description of the samples used is essential to facilitate understanding of the results (to give only a reference to these is not sufficient)

More detail has now been added on the Materials & Methods, and a Table (Table 1) with the % composition of each cream.

  • Line 62 and 76: I miss/suggest to add the source designation for the Sellmeier equation and coefficients the authors used

The online source (Wikipedia) of the Sellmeier coefficient has now been added (reference [14] of the revised manuscript).

  • Line 81-82: It is questionable to the interpretation of the results that only one person’s skin has been studied for the hydration effect. Because of it, I suggest a little modification in the Results and Discussion section (e.g.: preliminary results …)

Done. The word “preliminary” has been added in the Results & Discussion, to tally with the Method.

  • The differences of the RI values of the samples are small (Table 1). If the authors use different ingredients in the samples, it will also involve a change in RI values. But, if you change more than one parameter in a sample whether we can be sure that there is still a correlation between RI and the hydration effect.

A Table showing the composition of each cream has been added (Table 1 on revised manuscript) to support the discussion. Based on this, the water content & the type of humectant seem to be the determining factors on the hydration effect. The Discussion has been expanded.

  • Line 118: "determination of the IR of..." to be corrected: RI

Done

  • The relationship between the two parameters (RI and hydration effect %) could be illustrated in a figure which would improve the illustration of the results.

Please find this new Figure in the revised manuscript (Figure 1)

Reviewer 2 Report

The in vitro approach for screening the in vivo skin hydration of cosmetic products is an interesting method, although the presented paper looks to be an initial proof of concept research.

However, data from the in vivo study is very limited and it looks to be achieved from only one subject in triplicate. For a new method to be valid several subjects, that contributes to the variability of the human skin, should be evaluated, particularly if the in vitro method tries to replace the in vivo method. Authors should present more in vivo data in order to conclude about the use of the in vitro calculations.

Also, a more robust discussion about the influences of the different formulas on the data from both methods should be presented.

Author Response

We would like to thank the Reviewer for their constructive feedback.

The in vitro approach for screening the in vivo skin hydration of cosmetic products is an interesting method, although the presented paper looks to be an initial proof of concept research.

However, data from the in vivo study is very limited and it looks to be achieved from only one subject in triplicate. For a new method to be valid several subjects, that contributes to the variability of the human skin, should be evaluated, particularly if the in vitro method tries to replace the in vivo method. Authors should present more in vivo data in order to conclude about the use of the in vitro calculations.

This is a valid point which we have already accounted for in our manuscript. We clarify that the skin hydration study was “preliminary self-evaluation”. In the Conclusions we state that “Further studies should include a large scale user trial to validate this predictive tool”.

Also, a more robust discussion about the influences of the different formulas on the data from both methods should be presented.

A Table showing the composition of each cream has now been added (Table 1 on the revised manuscript). The Discussion has been expanded to include explanations on the influence of the formulas (Table 1) on the RI & hydration data.

Reviewer 3 Report

The proposal of using the measurement of the RI of a turbid sample such as O/W emulsions, which constitute the majority of moisturizing creams, as a parameter for predicting their moisturizing capacity seems interesting because of its simplicity, especially when the measurement is made indirectly from the transmittances and their relationship with the RI by means of the Sellmaier equation.

However, some questions should be taken into account:

In the statistical analysis, it is indicated that p<0.05 for significant difference, while in Results and Discussion it is indicated that p<0.001 has been considered as statistical significance. Please, clarify which is your refrence p-value.

Table 1: the selected wavelength shown in the seventh column is not expressed in nanometers but in micrometers, contrary to the individual values, expressed in nanometers. It is suggested to keep the units consistency.

Table 2: It is not clear either in Material and methods or in the table that the triplicate results of the skin hydration measurements were performed on the same individual or on different individuals. Particularly striking is the case of sample IA, in which the second hydration measurement is very low and not consistent with the other two measurements. Please, clarify better in the methodology. In addition, it should be interesting to show in this table the results of the t-test performed and the p value obtained for each cream.

Author Response

We would like to thank the Reviewer for their constructive feedback.

The proposal of using the measurement of the RI of a turbid sample such as O/W emulsions, which constitute the majority of moisturizing creams, as a parameter for predicting their moisturizing capacity seems interesting because of its simplicity, especially when the measurement is made indirectly from the transmittances and their relationship with the RI by means of the Sellmaier equation.

However, some questions should be taken into account:

In the statistical analysis, it is indicated that p<0.05 for significant difference, while in Results and Discussion it is indicated that p<0.001 has been considered as statistical significance. Please, clarify which is your refrence p-value.

Our reference p-value for statistical significance was p<0.05. In the Results and Discussion we highlight the p value rather than the threshold for significance. We have now corrected this sentence.

Table 1: the selected wavelength shown in the seventh column is not expressed in nanometers but in micrometers, contrary to the individual values, expressed in nanometers. It is suggested to keep the units consistency.

Thank you, there was the following typo which has now been corrected on the revised manuscript (Table 2 of revised paper):

The units of measured wavelength were in nm. We then converted the average wavelength to µm before using it on the Sellmeier equation to agree with the unit of the C coefficients which is µm2. All these units are now shown clearly on the revised manuscript.

Table 2: It is not clear either in Material and methods or in the table that the triplicate results of the skin hydration measurements were performed on the same individual or on different individuals. Particularly striking is the case of sample IA, in which the second hydration measurement is very low and not consistent with the other two measurements. Please, clarify better in the methodology. In addition, it should be interesting to show in this table the results of the t-test performed and the p value obtained for each cream.

The triplicate results were performed on the same individual, where the PhD researcher (first author of this manuscript) conducted this “preliminary self-evaluation of the hydrating effect of the creams” (line 100 of the revised manuscript) on herself, as explained in the methods. These preliminary results are interesting and merit follow-up larger scale trial to validate this predictive tool.

The p values have now been added on Table 3 of the revised manuscript.

Round 2

Reviewer 2 Report

Authors had review the text and include all the details about the formulations as requested. Also, a clearer discussion was included.
After this the paper is much clearer.
The inclusion of a statement about the preliminary state of the work is relevant as the number of in vivo measurements are very low and could jeopardize the innovation of the in vitro calculations.
An extensive and robust work comparing in vitro and in vivo data could be done in the future as this approach is very interesting and could be very relevant in screening phases of product development